# Implementing circularity measurements in industry 4.0-based manufacturing metrology using MQTT protocol and Open CV: A case study

**Yazid Saif**[1], **Yusri Yusof**[1], **Anika Zafiah M. Rus**[1]*, **Atef M. Ghaleb**[2]*, **Sobhi Mejjaouli**[2], **Sami Al-Alimi**[1]*, **Djamal Hissein Didane**[3], **Kamran Latif**[4], **Aini Zuhra Abdul Kadir**[5], **Hamood Alshalabi**[6], **Safwan Sadeq**[7]

1 Faculty of Mechanical and Manufacturing Engineering, Universiti Tun Hussein Onn Malaysia, Batu Pahat, Malaysia, 2 Department of Industrial Engineering, College of Engineering Alfaisal University, Riyadh, Saudi Arabia, 3 Center for Energy and Industrial Environment Studies, Universiti Tun Hussein Onn Malaysia, Batu Pahat, Malaysia, 4 Department of Engineering Technology, Faculty of Technical and Vocational, Universiti Pendidkan Sultan Idris (UPSI), Tanjung Malim, Malaysia, 5 School of Mechanical Engineering, Faculty of Engineering, Universiti Tecknologi Malaysia (UTM), Johor Bahru, Malaysia, 6 Sana'a University, Sana'a, Yemen, 7 School of Engineering, Design and Built Environment Western Sydney University Penrith, Penrith, NSW, Australia

* zafiah@uthm.edu.my (AZMR); aghaleb@alfaisal.edu (AMG); samiabdo@uthm.edu.my (SAA)

**Data Availability Statement:** All relevant data are within the manuscript and its Supporting information files. -Full access to the code for

## Abstract

In the context of Industry 4.0, manufacturing metrology is crucial for inspecting and measuring machines. The Internet of Things (IoT) technology enables seamless communication between advanced industrial devices through local and cloud computing servers. This study investigates the use of the MQTT protocol to enhance the performance of circularity measurement data transmission between cloud servers and round-hole data sources through Open CV. Accurate inspection of circular characteristics, particularly roundness errors, is vital for lubricant distribution, assemblies, and rotational force innovation. Circularity measurement techniques employ algorithms like the minimal zone circle tolerance algorithm. Vision inspection systems, utilizing image processing techniques, can promptly and accurately detect quality concerns by analyzing the model's surface through circular dimension analysis. This involves sending the model's image to a computer, which employs techniques such as Hough Transform, Edge Detection, and Contour Analysis to identify circular features and extract relevant parameters. This method is utilized in the camera industry and component assembly. To assess the performance, a comparative experiment was conducted between the non-contact-based 3SMVI system and the contact-based CMM system widely used in various industries for roundness evaluation. The CMM technique is known for its high precision but is time-consuming. Experimental results indicated a variation of 5 to 9.6 micrometers between the two methods. It is suggested that using a high-resolution camera and appropriate lighting conditions can further enhance result precision.

measuring data is available in Appendices A and B, included as Supporting information in a supplementary file. I want to emphasize that all data, without exception, is comprehensively documented and presented within the manuscript itself.

**Funding:** This research was supported by the Ministry of Higher Education (MOHE) through the Fundamental Research Grant Scheme (FRGS/1/2020/STG01/UTHM/02/2).

**Competing interests:** The authors have declared that no competing interests exist.

**Abbreviations:** *3D*, Three-dimension model; *3SMVI*, Smart System based on interpreted STEP-NC for Machine Vision Inspection; *ACO*, Colony Optimization; *ANSI*, American National Standards Institute; *C*, Center of this circle; *CAIP*, Computer-Aided Inspection Planning; *CLIM*, Closed-loop inspection manufacturing; *CMM*, Coordinate Measuring Machine; *CMOS*, Complementary metal-oxide semiconductor; *CNC*, Computer numerical control; *e MZC*, Error based Minimum Zone radial circles; *EO*, Engineering Ontology; *FTP*, File Transfer Protocol; *GA*, General Assembly Simulation; *GD&T*, Geometric Dimensioning and Tolerance; *I 4.0*, The fourth Industry Revolution; *IoT*, Internet of Things; *ISO*, International Standard Organization; *LED*, Light-Emitting Diodes; *MQTT*, Message Queuing Telemetry Transport; *MTD*, Metrology for the Digitalization; *MZC*, Minimum Zone for Radial Circles; *OLP*, Off-Line Programming System; *OOR*, Out-Of-the-Round; *Open CV*, Open Computer Vision; *QM*, Quality MITUTOYO Measure; *RealVNC*, Real Visual network center; *RGB*, Three Channel of color Red, Green and Blue; *Ri*, Radius inner; *RL*, Lower radius; *RU*, Upper radius; *STEP-NC*, The standard for the Exchange of Product Model Data for Numerical Control.

# Introduction

Recent advancements and automation in metrological systems have led to substantial improvements in measurement speed and accuracy. Metrology plays a crucial role in the manufacturing industry by ensuring quality control. It involves the measurement and inspection of machine parts, components, and assemblies to ensure they meet the required specifications. As manufacturing continues to evolve with automation, personalized production, and global expansion, its significance remains paramount. Vision inspection systems are extensively employed in industrial manufacturing environments to identify and address product defects. These systems examine product surface properties and identify potential defects using image processing methods. By using vision inspection systems, manufacturers can quickly and accurately detect defects, allowing them to take corrective action and prevent defective products from reaching consumers.

As manufacturing industries adopt advanced technologies and methods, they play a leading role in the implementation of 4.0 industry. This revolutionized the interconnection of various elements in the industry, allowing manufacturers to maintain competitive advantages in a world that is evolving rapidly and adapting quickly to customers' needs. Manufacturing measurement is an important aspect of advanced manufacturing systems, which is rooted in Industry 4.0, as it covers the evaluation and evaluation of almost all machining components in specialized measurement systems. Thus, according to Berthold and Imkamp [1], Industry 4.0 provides metrology support in three key areas within the manufacturing sector: (a) smart supply chains with cyber-physical metrological traceability, (b) smart manufacturing incorporating cyber-physical metrology in manufacturing processes, and (c) smart products employing intelligent metrology.

As part of Industry 4.0, a smart metrology inspection planning system based on Coordinate Measuring Machines (CMM) has been developed. The system is structured around three artificial intelligence (AI) techniques, namely engineering ontology (EO), ant colony optimization (ACO), and general assembly simulation (GA) [2]. Metrological Technology Board (PTB) intend to focus on digital transformation, large-scale data analysis, and digitalization communication networks. Therefore, next stage of its Metrology for the Digitalization (MTD) project, innovative Metrology uses artificial intelligence (AI) technologies to create an updated metrological function for making critical decisions. Based on Lazzari [3]. A novel metrological model called the cyber-physical manufacturing metrology model (CP3M) was created specifically for the Industry 4.0 framework. Enhancements in inspection methodologies for measuring instruments rely on robust software assistance for various task categories, such as tolerances. The authors' prior research focused on the development of an intelligent system for organizing PMP (Product and Manufacturing Process) inspections using a Coordinate Measuring Machine (CMM) [4, 5], a precise technique was created to determine an appropriate measuring path for a single. The implementation of consistent inspection for such devices is a unique challenge. The complexity of prismatic components, the inspection planner's expertise and experience, and their perception all contribute to determining it. As stated by [6] key challenges encountered when defining the product and its measurements include the need to incorporate geometric dimensioning and tolerance (GD&T) data into CAD models, handling nonstandard GD&T information, and the lack of computer-readable (interpretive) and standardized definitions for measuring equipment capabilities, configurations, performances, and, etc.

According to the tentative review-based inspection of Closed-loop inspection manufacturing (CLIM) and computer-aided inspection planning (CAIP) has grown into a size approach that can examine industrial fashion in the industrial area [7]. However, CAIP is rushed in the

device and will become a critical stage in the production system. As a result, a thorough literature review aims to evaluate current inspection studies and the use of CLIM in CNC machines. The review focuses on CAIP and CLIM, system-based technology, and the implementation of STEP/STEP-NC. In addition, the study provides a comprehensive taxonomy and defines advanced features of this emerging technology [7, 8]. Thus, the 3SMVI is a development of a smart system for machine vision inspection with STEP-NC environmental based on cyberphysical which capable of decision-making at a given moment [9]. The importance is placed on generating the ideal measurement sensor path as a whole section of prismatic parts inspection. It aims to eliminate intuition, presentation of knowledge, reuse and share of knowledge. Overall, computer vision technology has revolutionized the manufacturing industry by improving the quality control process, reducing costs, and increasing productivity.

This study aims to develop an IoT-based machine-vision system that can effectively and efficiently detect circular holes in surface characteristics during milling machine operations. The system implements the MZC problem solution technique to improve roundness measurement. A workpiece with circular features has been embedded in the IoT-enabled vision machine system. The vision machine includes a cloud server with a wireless router board and an intelligent camera sensor built into the Raspberry Pi 4. The primary objective of this study is to measure the surface characteristics of a DELRIN workpiece that has been milled using a non-contact inspection approach that employs an in-situ CMOS camera for circle hole inspection. The focus of the study is on the development of a smart system called 3SMVI (interpreted STEP-NC file for machine vision inspection) for offline measurement and real-time inspection using a coordinate measuring machine (CMM). This article is organized: **Section 2** addressed the related research for machine vision systems; **Section 3** defines Methodology, Setup and Procedure, The Method for Determining the Shape of a Circler Form, the proposed experimental setup and procedure, and the implementation of 3SMVI through MQTT **Section 4** discusses an Image processing implementation Algorithms. **Sections 5 and 6** represent the advancement of Vision System with Computer Algorithm and utilize the Coordinate measuring machine (CMM) method, along with the outcomes. Results and discussion are addressed in **section 7**. Finally, the authors' conclusion and future trends are outlined in **Section 8**.

## Previous studies in vision inspection system

The manufacturing industry has long sought to develop a system that could observe and describe whatever it observed. The key reasons a computer should not necessarily seek to duplicate the human visual system are speed, cheap cost, and significant dependability. This idea became known as machine vision in the industrial sector [10]. Computer vision that is employed in manufacturing or beyond practical uses is known as machine vision. It is utilized to execute a particular feature or performance based on the vision system on picture analysis. As a result, automated extraction of vital information from digital photographs to measure operations or examine produced manufactured. Computer vision has become increasingly important in regulated industrial applications over time, and it now offers innovative approaches for the sector of industrial automation. Computer vision applications have greatly assisted a number of industries, including the manufacturing of electronic components, including more effective manufacturing processes [11, 12], fabric defect and detection inspection [12], automated food sorting system or techniques [13–15], flexible automated assembly systems [16], incorporating circuits inspection [17]. Machine vision systems are often classified into four categories [18]: automatic visual inspection systems, parts recognition, robotic supervision, and process control.

## Machine vision-based defect detection

In recent years, there has been an increase in interest in implementing computer vision systems in industrial workplaces to detect product defects. These systems use various image processing techniques to analyze the features and surface of the model, providing valuable information to the experts. Advanced industrial systems necessarily involve ever-improving product performance and improved product quality during the production process [19–21]. Defects, such as scratches, imperfections, or holes on the products surface, on the other hand, have an adverse influence mainly on the products aesthetics and user comfort, but also its performance [22–25]. Therefore, Defect detection is an efficient strategy for minimising the environmental effects of product defects [26–28]. Machine vision systems have the potential to assist in various tasks within industrial processes [29–31].

An industrial vision inspection system typically comprises essential components such as optical illumination, image acquisition, image processing, and defect detection. These components work together to improve the inspection and detection of defects in industrial settings [32, 33]. This research aims to create an intelligent system that automatically interprets and measures surface characteristics according to Example 1 Part 21 of the ISO 14649 standard, utilizing machine vision inspection (3SMVI). The 3SMVI system will be developed using a camera and lighting system on a milling machine. The primary purpose of this system is to enable the non-contact inspection of circular holes, making it easier to measure surface features on milled workpieces [9, 34]. The 3SMVI system has become a commonly utilized tool for surface measurement and inspection. This article investigates the growing utilization of camera systems to verify the precision of surface features in both design and measurement processes. It also discusses the challenges of measuring roundness in circles using vision and validates the system based on IoT.

## Materials and methods

The methodology used for developing the vision system of 3SMVI integrated with IoT, which contrasts current approaches. The fundamental concept and architecture of the 3SMVI system are outlined, and the advancement of the visual system based on the system structure is described. The objective of this research was to develop intelligent systems capable of conducting machine vision inspections in IoT environments by utilizing measurement data. These systems were implemented using an interpreted file derived from ISO 14649 Example 1 Part 21. The system developed facilitates the conventional 3-axis CNC milling machine with a complete vision system architecture.

### Method for determining the shape of a circler

The surface of any engineering component provides insights into its structural characteristics, which primarily include three aspects: form errors, roughness, and waviness. Straightness, flatness, and circularity (roundness) are critical requirements that need to be satisfied during the assembly and mating of various parts. Out-of-round (OOR) refers to the difference between the actual radius of a circle and the measured radius at each location. Roundness measurement, as per the ANSI Y 14.5M (1994) standard, commonly involves four reference circles: The Maximum Inscribed Circle (MIC), the Minimum Circumscribed Circle (MCC), and the Minimum Zone for Circles (MZC). The recommended method for measuring roundness, according to established standards, is the Minimum Zone for Circles (MZC). This method generates more consistent results compared to other approaches and aligns with the standards set by ANSI and ISO. For roundness measurement, four reference circles are globally recognised in (ANSI Y 14.5M, 1994), according to [35–37]. ASME Y14.5M-1994 and ISO 1001,

roundness assessments based on MZC include identifying primary circles that include all profile points and ensuring the minimum radiation difference between these circles. Therefore, this section will focus on the roundness approach, which follows the minimum radius circle area (I) of Section 1.

**Minimum zone circles.** The roundness error calculation method involves comparing two circles. (Fig 1) illustrates the process, where an outer circle is drawn to encompass the entire circularity profile, and an inner circle is drawn within the roundness profile to match the shape. The minimum radial distance between these two concentric circles, known as the circumcised and inscribed circles, is measured. Subsequently, the center of the minimum zone circles, which corresponds to the center of the measured form, is determined. Because the radius of the minimal zone circles varies from one another, the profile is rounded [38]. Thus, the minimal radial separation is the roundness inaccuracy of the MZC approach.

$$\left\{\begin{array}{l} R_1 - R_2 \\ \sqrt{(xi - xc)^2 + (yi - yc)^2} \leq R_1 \\ \sqrt{(xi - xc)^2 + (yi - yc)^2} \geq R_2, i = 1, 2, 3, \ldots n \\ R_1 - R_2 \geq 0 \\ R_L \leq R \leq R_U; \\ x_L \leq x_c \leq x_U; \\ y_L \leq y_c \leq y_U \end{array}\right\} \qquad (1)$$

The roundness error, which represents the minimum radial separation, is determined by the difference between the radii of C1 (indicated as R1) and C2 (indicated as R2), which share the same center coordinates (x, y). It is mathematically formulated as

$$e_{MZC} = R_{max} - R_{min} \qquad (2)$$

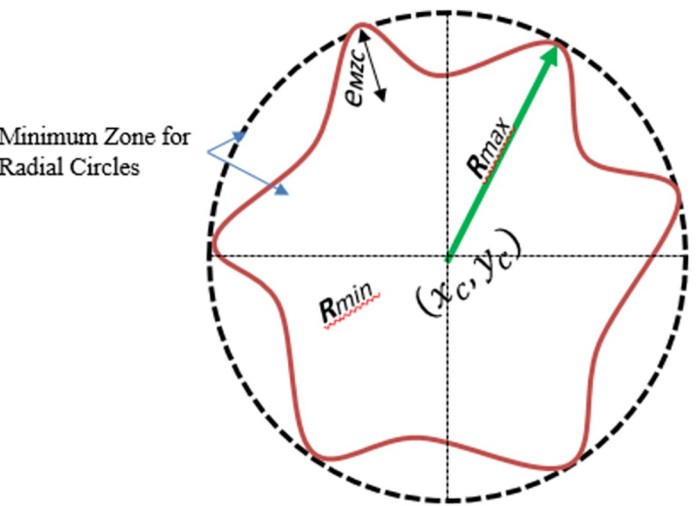

**Fig 1. Principle of algorithm for Minimum Zone for Radial Circle (MZC).**

## Experimental setup and procedure

The investigational configuration comprises of a combination of hardware and software that enables non-contact roundness measurement through a vision system. To ensure accurate measurements of the workpiece, LED lighting is utilized to eliminate shadowing that may result in the loss of pixels during capturing. Additionally, the measuring system involves a servomotor that rotates the test model or workpiece.

**Case study.** The vision system design was specifically adapted to the 3-axis ProLIGHT 1000 CNC milling machine and adheres to the guide lines contained in the ISO 14649 Part 21 file. The aim of choosing the model based STEP-NC due to the growing global manufacturing industry, ISO 10303 and ISO 14649standards, known as STEP and STEP-NC, have been developed to improve interoperability and streamline the planning and manufacturing of discrete components, especially in milling and turning machines centers, that could meet the dynamic demands of industrial production. Through extending the reference model that complies with STEP-NC File 21 and assigning priority to the identification of inspection holes and surface areas in the CAD model, it can improve the overall precision and quality assurance paradigm of CNC milling machines. Thus, the CAD model correctly designed by CATIA software using the surface shape of the model, particularly in the four holes that require specifically inspection the surface area. A certain circle is strategically used to cover a considerable portion of the surface of the model. This technique facilitates the accuracy of measurements of surface properties. However, the extended model enhanced the version of the ISO 14649 Part 21 File, which wisely includes four holes and additional pockets, as shown in (Fig 2). The following example expands deeply into the specifics of the DELRIN prismatic workpiece's machining parameters and emphasizes the presence of round holes with diameters of 22mm. The operational parameters are set carefully: the spindle speed is calibrated at 2000 rpm, the feed rate is harmonized to 0.1 mm/rev, the cutting depth is firmly set at 20 mm, and the cutting speed is judiciously maintained at 250 m/s.

The guidelines for developing the STEP-NC code of standardization for milling CNC machines are exactly what is identified in this machine's parameters. However, this data could be applied by a PROLIGHT 100 CNC milling machine to drill the four holes in the Delrin

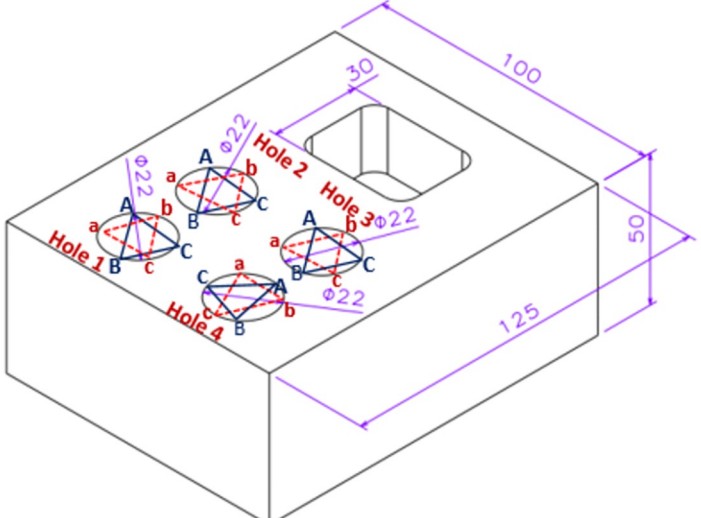

**Fig 2. Part design of Delrin mild workpiece.**

**Table 1. Characteristics of parameters and specifications for mild Delrin model.**

| Machine characteristics | Parameter | Specification |
|---|---|---|
| Machine part | Workpiece | Delrin |
| | Size | 125x100x50mm |
| Machining | Feed rate | 0.1mm |
| | Spindle speed | 2500rpm |
| | Depth of cut | 22mm |
| | Four Circle diameter | 22mm |
| | Cutting speed | 250m/s |
| Cutting tool | Material type | High-speed steel |
| | Diameter | 0.6mm |
| | Number of flutes | 2 |
| | Tool type | New tool |
| CMM measurement | Number of axes | Three axis's X, Y&Z |
| | Machine type | MITUTOYO QM- 353 |
| | Resolution | 0.00002 |
| | Accuracy | Highly |
| | Flexibility | Moderate |

workpiece in accordance with the standards to obtain better surface features after the product's manufacturing is achieved. Table 1 summarises the key aspects of the machining part, the supporting machining parameters, the characteristics of the cutting tool, and the measuring machine specifications.

**3SMVI system coordination and calibration.** The 3SMVI interface program calculates pixel size in the x and y-axis based on the actual size of the extanded model based on STEP-NC Example 1 Part21 file. It calculates the maximum diameter of the outer edge of the circle, examines the inner contour's edge pixels, calculates a calibration element in the x-direction ($CFx$), and generates a calibration element in the y-direction ($CFy$). The program multiplies all edge pixels' *x and y* coordinates by these values. This method can be used to derive a fundamental roundness circle error. Therefore, the coordinate system, defined by a reference point origin and angle within an image, allows for a search area based on the object or model's characteristics. Pattern matching locates surface features, creating an accurate coordinate system. A region of interest (RoI) is defined for analysis. Moreover, the camera system was calibrated using a linear model that ignores lens distortion, focusing on the relationship between the three-dimensional geometry of a point on the workpiece's surface and its equivalent point in the image, with four coordinate systems for images and pixels.

In Industry 4.0 environments characterized by heterogeneity, the proposed MQTT solution plays a pivotal role in tackling interoperability challenges. MQTT's flexibility and protocol-agnostic nature make it adept at handling various sensor types, data formats, and communication protocols. By decoupling producers and consumers through its publish-subscribe model, MQTT allows different sensors to communicate seamlessly, reducing the need for extensive protocol translation or adaptation. Section 3.3 elaborates on the hardware and software development, dedicating subsections 3.3.1 and 3.3.2 to hardware and software design, respectively. Furthermore, section 3.4 the MQTT Installation Python and section 3.5 outlines the communication aspect of 3SMVI using MQTT in IoT applications., and specifically discusses the combination process involving the Cloud server and the MQTT communication protocol.

## Hardware and software of development system

**3SMVI hardware development.** The 3SMVI system is a hardware-based vision inspection system that incorporates various components, including a digital webcam, power supply, internet adapter, lighting system, step motor, personal computer, and driver. Experimental tests conducted at the Advanced Material and Manufacturing Center Lab (AMMC) in UTHM demonstrated the webcam's capability to capture surface feature images of up to 5 megapixels with a resolution of 640x480. Based on the hardware deteiles adressed as the folowing:

- A digital webcam utilized in this study with specification CMOS sensor, Global shutter Resolution: 640X480MJPEG 30fps (5 MP).

- A power supply set in a 3SMVI system, also called a power distribution box, makes it easier to manage power for numerous cameras at one location. With some having manual or automatic input voltage selection, it converts AC to low-voltage DC power for computer components.

- Network adapters enable computers and other devices to communicate with the Internet or other local area networks (LANs) in 3SMVI. Wireless network adapters for laptops and tablets convert computer signals into radio waves.

- Implementing the lighting system for the 3SMVI platform was challenging due to the significant impact it has on the system's performance and quality, necessitating standard steps to improve the contrast images captured.

- The stepper motor has the ability to divide a single full rotation into a series of smaller (and essentially identical) rotational components. It could be utilized to instruct the stepper engine to rotate or move at a specific angle or degree for practical purposes.

- The personal computer was an Intel(R) workstation with a Core (TM) i7 processor and 512 MB of storage that the 3SMVI used.

- A driver has emerged as a component of software that facilitates communication between a device and the operating system. Consider a scenario where an application needs to read data from a device.

The system is designed to support IoT and integrates elements such as FileZilla, Raspberry Pi, camera interface, LED lighting, and Wi-Fi router. Proper configuration of lighting is essential to ensure high-quality image capture for inspection purposes. With its hardware and software components, the system enhances monitoring, enables measurement in specific areas of interest, and facilitates connectivity with cloud servers, making it well-suited for industrial 4.0 applications.

- The FileZilla component, renowned for its adeptness in secure file transfer, assumes a pivotal role in this ecosystem. By enabling seamless and secure data exchange, it lays the foundation for effective communication within the IoT network.

- Furthermore, the inclusion of the 'Raspberry Pi,' a versatile microcomputer, serves as the backbone of computation and connectivity. Its compact yet robust architecture empowers the system with processing capabilities, rendering it adaptable to a plethora of tasks.

- The 'camera interface' element signifies the integration of visual perception into the system. Through this component, the system gains the ability to capture and interpret visual data, a crucial facet for applications such as surveillance, analysis, and real-time monitoring.

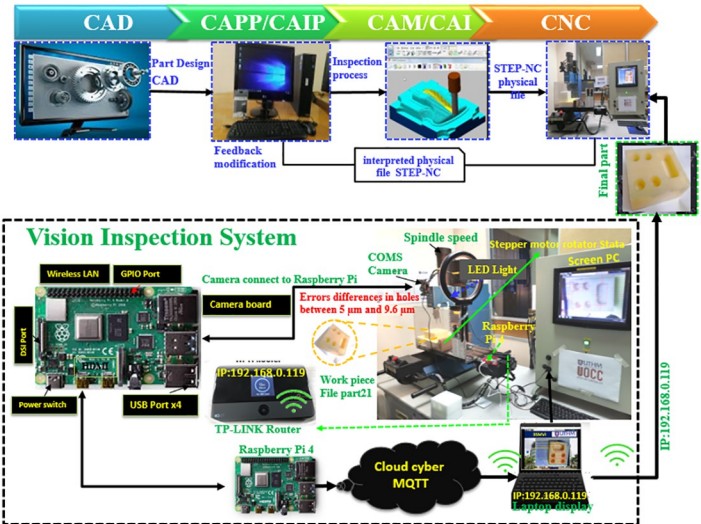

**Fig 3. The implementation of 3SMVI system.**

- The LED lighting emerges as more than just illumination; it assumes a role in signaling, status indication, and even data transmission. Its controllable and programmable nature makes it a versatile tool in conveying information within the IoT framework.

- The Wi-Fi router stands as the linchpin connecting all these elements cohesively. By providing wireless connectivity, it facilitates the seamless exchange of data between various components, fostering the harmonious functioning of the IoT ecosystem.

One application of the 3SMVI system involves measuring the roundness of holes in DELRIN workpieces created using a CNC milling machine. The system rotates the workpiece in the Z-direction using the spindle module to inspect the circularity of the holes. The research paper describes the rotational orientations of the hole's rotation as observed from the +Z-direction. The top vision module, depicted in (Fig 3), consists of an Imaging Source DMK41BUC02 complementary metal-oxide-semiconductor (CMOS) camera and a vertically positioned LED light ring illuminator. These components are mounted on a gantry-type frame, with the camera axis aligned along the Z-direction of the 3SMVI system. Fine adjustments can be made to the position and focus of the top CMOS camera in the Y- and Z-directions. (Fig 3) illustrates the experimental setup of a cloud-based MQTT system for the 3SMVI prototype, developed based on Part 21 of the ISO 14649 standard and utilizing the INTELITEK PROLIGHT 3-axis milling machine. The 3SMVI system was specifically designed to enable accurate inspection of various models and workpiece components.

**Software development for 3SMVI.** The 3SMVI software is designed to enable the development of a smart system for machine vision inspection using the STEP-NC environment and an IoT application. It consists of various components that promote integration and interoperability among devices.

The vision inspection system comprises four main parts: a data diffusion mechanism, operational modules, a data model, and a computational intelligence algorithm, as depicted in (Fig 3). OpenCV was utilized to develop four inspection modules, allowing for the detection and measurement of surface feature dimensions in the model. These modules were implemented within the STEP-NC environment, and a software-based IoT architecture was integrated to ensure comprehensive inspection functionality. The four modules include the measuring

module, image processing module, image analysis module, and measured identification module. The machine vision component of the software utilizes the Open CV library, implemented in Python and compatible with the Raspberry Pi 4 microcontroller board. All modules are fully operational, providing visualization of the inspection process and delivering the required geometric dimensions.

## MQTT Installation python

The MQTT protocol is a machine-to-machine (M2M) IoT connectivity protocol for sending data from IoT devices to cloud service providers like Microsoft Azure. It's ideal for remote interactions with a small code footprint and higher network bandwidth. Python swiftly uses the MQTT route essence, with two computers as clients and one as a broker. One of our computers serves as a broker. Identify an IP as a module and publisher for sending information to the MQTT broker. However, a publisher was a computer. The MQTT python program would seem to be a light and robust process to install and run. When accessing MQTT in Python, it may employ the popular library "paho-mqtt." The paho-mqtt library is used to install and use MQTT in Python. The folowing steps addressed below:

- Python Installasion: make sure that the downloading python to the system form official website.

- Install the "paho-mqtt" using pip from package manager to install library of paho-mqtt and openthe termenal or command promot to run into the system as folowing command: "pip install paho-mqtt".

- Start to write the Python code and use the MQTT functionality to publish data and subscripe it.

- By relpacing the proker address and port based on IP and port number of the MQTT broker.

- Then, run the code and save it. It will connect to the MQTT broker, subscribe to the specified topic, publish a message, and print received messages.

## Application for 3SMVI within MQTT-based communication with Internet of Things

IBM established the MQTT messaging protocol for consumer-oriented applications. Its utilization of it has grown significantly over time in a variety of industries, including office automation, home automation, and health services.

MQTT emerges as a favorable choice for facilitating machine-to-machine communication, presenting notable distinctions from other protocols such as OPC-UA, MT Connect, AMQP, and CoAP [8]. CoAP was constructed using UDP or HTTP as the primary transport protocol [39, 40], but MQTT depends on TCP or IP. MQTT boasts several distinct advantages, including its capacity for streamlined data transfer, minimized bandwidth usage, and adept handling of scenarios characterized by high latency or unstable network connections. Research conducted by Rocha et al. [41] has demonstrated the applicability of MQTT (Message Queuing Telemetry Transport) for low-power and low-latency applications, especially on wireless devices like smartphones. MQTT is known for its lightweight nature and cost-effectiveness, utilizing a central broker to facilitate communication between clients. Moreover, MQTT exhibits versatility by accommodating both wired and wireless connections, showcasing efficiency through its minimal memory requirements. For applications demanding heightened

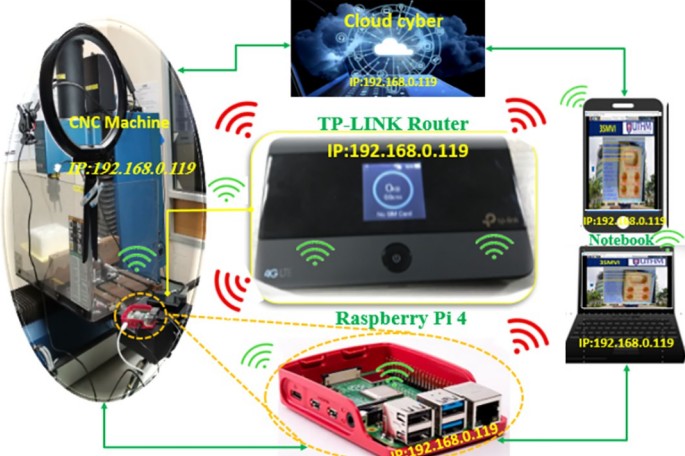

**Fig 4. A network graphic represents IP addresses.**

precision and reliability, MQTT offers a Quality of Service parameter. This protocol enjoys widespread support across prominent IoT platforms, counting among its proponents industry giants such as IBM, Microsoft, Amazon, and LabVIEW.

In the Python programming language, a Paho-MQTT client library has been developed, enabling the testing of client-to-MQTT functionalities on the Raspberry Pi 4. Clients can subscribe to specific topics and communicate through these topics without the need for any system configurations when connected to the broker's IP address. In the context of this study, the Paho-MQTT library is employed to publish data from each camera sensor, while the Raspberry Pi acts as a wireless communication broker. Internet connectivity is established by connecting the Raspberry Pi to the internet via a Wi-Fi adapter, with a wireless router being utilized to provide internet access. The network system depicted in (Fig 4) allows devices such as the Raspberry Pi, laptop, computer, and smartphone to access the same IP address and communicate with each other. MQTT specification outlines control packets and codes; extended codes in **Appendix A in** S1 File.

The IoT-based inspection software develops three main layers: perception layer, networking layer and application layer. Therefore, the IoT architecture-based vision inspection device enables the CNC operation model to be sensed by the camera sensor, transmitted to the cloud-based MQTT server, and sent back to open CNC PCs through seamless Integration and communication.

**The Perception Layer** is the intelligent device layer or senses or activators [42]. It senses and gathers information around it and subsequently transfers it to the communication layer. In this study, there are three categories of device, like the following:

- The camera sensor was the device for sensing and capturing the surface feature of the model produced by the CNC milling. In order to capture an accurate product, the LED illumination can reduce the shadow of the model.

- Paho-MQTT class provides the Paho-MQTT to import Python code from the library, which must have the line of code "Import paho.mqtt.client as mqtt." This client must have a unique ID and the mqtt.Client() method can be used.

- Raspberry Pi was a broker that enables the server to accept information from the Wireless Adapter and to transfer it to the Web browser from the RealVNC server.

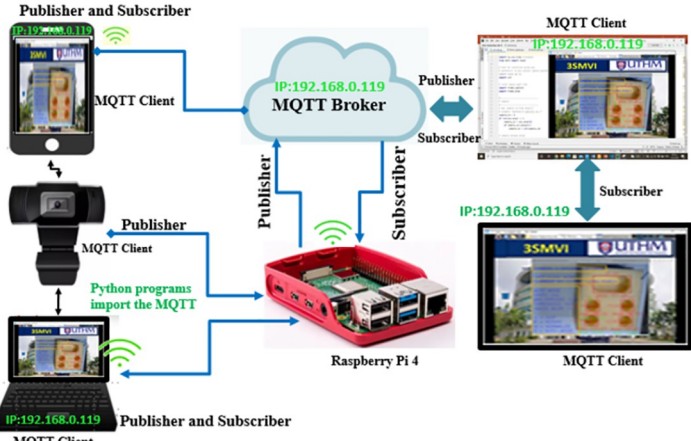

**Fig 5. The MQTT test broker with Python program.**

**The Network Layer** ensures that data from the camera sensor to the application layer are sent and processed via optical sensors' connectivity and network servers. In the current study, communication between the camera sensor, the software and the Raspberry Pi (network device) capable of connecting and verifying the MQTT test broker, and finally to the application layer, was performed as shown in (Fig 5). Three sections were involved in the MQTT information chain:

- Publisher was where it would generate the capturing images and send information to the MQTT broker. The camera sensor gathers images of the model's surface data as an information generator. The camera sensors were connected with a Raspberry Pi that serves as a gateway.

- The MQTT Broker was the critical point of communication and was responsible for sending all messages between senders and eligible receivers. The cloud-based MQTT broker operates as a gateway.

- Subscribing to topics requires a special method called a "callback" and using the function below "ourClient.subscribe("AC_unit")".

**The Application Layer** can provide the end-user or the end platform with relevant information about an intelligent device or camera sensor. The application layer was separated into two layers in the current study: the service and interface layers. Hence, the service layer offers the information as a visualization, while the interface layer is intended to interface the information. This study includes an Open CV in the service layer and a Real Virtual Network Computing (RealVNC) viewer in the interface layer. The interface layer acts as an interface to inspect the perspective, emphasizing roundness holes' information transferred in real-time from the open cv platform to the cloud server using an Open CV platform-operated CNC milling PC.

## Image processing implementation and algorithms

Pre-processing has the primary objective of enhancing specific texture features and eliminating unexpected distortions to improve image data. The objective of this stage is to prepare the image data for subsequent development phases. The visual sensor image may contain various

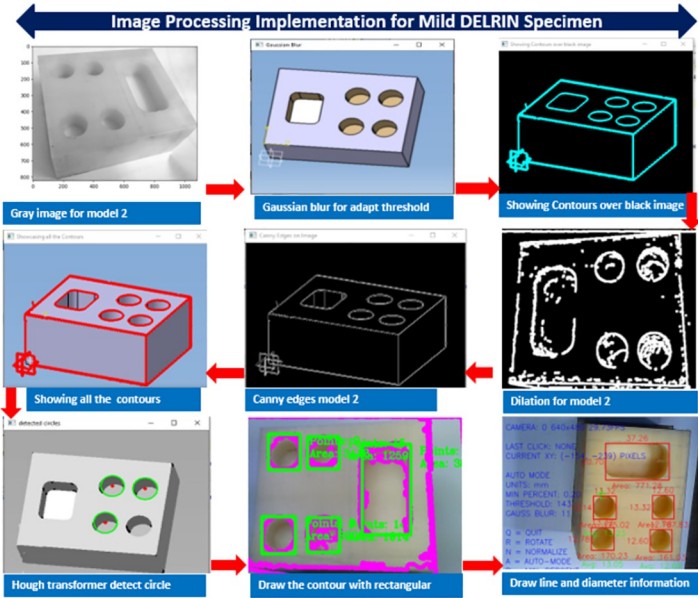

**Fig 6. Implementation of image processing technique.**

imperfections such as noise, glare, or blurriness. (Fig 6) illustrates a sample image obtained from the vision system. To ensure smooth progression through the operational modules, the inspection system utilizes algorithms for image processing, image analysis, computational intelligence, and measurement identification. For example, when dealing with a grayscale fingerprint image, it needs to be converted to pure black and white without any background interference for further processing. Additionally, image calibration is performed by adjusting the RGB settings in the camera's software to align the colored measurements with the corresponding reference image using a color tester image. Hence, the extended codes are accordingly found in **Appendix B in** S1 File for image processing.

## Automated data acquisition from 3SMVI vision

This roundness measurement technique enables automated data acquisition by utilizing a machine vision system. The workpiece is rotated at specific intervals, and a high-end Webcam camera, positioned directly in front of the object at a 90-degree angle, captures images. During each measurement cycle, 330 images are taken of the target area. The workpiece rotation is achieved using a 2-phase stepper motor controlled by an Arduino board, which offers enhanced resolution due to its smaller step angle and sufficient torque. The application establishes connections with MQTT clients through internet connectivity, using the client's IP address as (IP:192.168.0.119), and port number as (21) to transffer the data through visual network computing (VNC). The VNC module serves as the primary interface for the machine vision platform, displaying image files. (Table 2) presents the surface feature of a hole pocket, including four circle hole diameter values (22 mm each), generated at various angles ranging from 0 degrees to 360 degrees within the 3SMVI vision system. The standard circular error in micrometers for each hole was also calculated.

## Cyber-physical systems in an IoT environment, for 3SMVI system

After establishing the connection, the VNC module is responsible for displaying the image file, while Module 1 acts as the primary interface of the 3SMVI system. In (Fig 6), it can be

**Table 2. Diameter values from vision system for mild Delrin workpiece.**

| Image. No | Angle in degree | Hole 1 Dia. 22 mm | Hole 2 Dia. 22 mm | Hole 3 Dia. 22 mm | Hole 4 Dia. 22mm |
|-----------|-----------------|-------------------|-------------------|-------------------|------------------|
| 1 | 0˚ | 22.2693 | 22.1297 | 21.9897 | 21.9432 |
| 2 | 31˚ | 22.2473 | 22.1209 | 21.9797 | 21.9392 |
| 3 | 62˚ | 22.2695 | 22.1211 | 21.9808 | 21.9421 |
| 4 | 93˚ | 22.2581 | 22.1231 | 21.9801 | 21.9399 |
| 5 | 124˚ | 22.1902 | 22.1223 | 21.9808 | 21.9447 |
| 6 | 155˚ | 22.2491 | 22.1219 | 21.9891 | 21.9452 |
| 7 | 186˚ | 22.2506 | 22.1191 | 21.9756 | 21.9391 |
| 8 | 217˚ | 22.2479 | 22.1276 | 21.9822 | 21.9389 |
| 9 | 248˚ | 22.2459 | 22.1216 | 21.9814 | 21.9385 |
| 10 | 279˚ | 22.2397 | 22.1203 | 21.9823 | 21.9432 |
| 11 | 310˚ | 22.2689 | 22.1201 | 21.9799 | 21.9422 |
| 12 | 360˚ | 22.2676 | 22.1209 | 21.9816 | 21.9498 |

observed that Module 1 showcases a comprehensive description of the machine vision platform and the corresponding image file. The design of the system prioritizes the organization and transmission of logical data sets to various clients, encompassing smart devices such as tablets, smartphones, and workstation PCs. This is achieved through the utilization of both the visual network and MQTT broker. By adopting this structure, the system can effectively capture and articulate the cyber-physical characteristics involved in machine vision inspection, thereby facilitating a detailed analysis of the inspection process.

Module 2, coded in the Python programming language, is in charge of identifying and studying circular shapes present on the model's surface attributes. Module 3 introduces the machine vision inspection interface, which exhibits the precise positioning of the model along the X and Y axes. It also defines 'resolution' as 'pixels per inch' to ensure optical clarity. Module 4 highlights the auto-mode frame configuration within the 3SMVI system. This mode empowers the system to autonomously execute and scrutinize image data associated with a particular surface attribute. It accomplishes this through a modified script reliant on OpenCV. The measurements acquired through this analysis are presented in millimeters (mm), with a minimum threshold established at 0.2. The threshold parameter regulates the image's level of clarity, as demonstrated in Fig 7. This frame pattern underscores the system's adaptability, granting users the capability to rotate the model or halt the analysis by pressing the 'Q' key.

## The 3SMVI integration

This research endeavor aimed to create an innovative solution by developing an intelligent system that utilizes interpreted STEP-NC files for conducting machine vision inspections within an Internet of Things (IoT) framework. The primary focus of this endeavor was to seamlessly merge the processes of machining and inspecting the roundness of milled workpieces. This integration was achieved by implementing the system on a PC-based CNC machine, which was connected to a cloud platform. The entire operation was facilitated through the utilization of Open CV technology, leveraging the novel approach of IoT.

In this setup, each individual device and machine vision module was interconnected through a shared IP address as (IP:192.168.0.119), enabling wireless communication via the internet. This architecture capitalizes on the IoT paradigm, facilitating the integration of inspection procedures and enabling effective communication through a central server

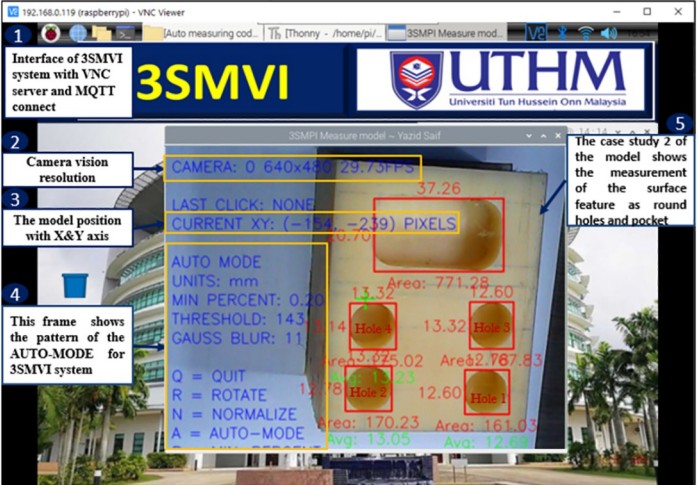

**Fig 7. The 3SMVI system's interface for obtaining measurements.**

broker. Particularly, the IoT approach presented a unique opportunity to establish a cohesive link between the various components involved. A fundamental aspect of this approach involved the utilization of interpreted STEP-NC files derived from the CNC machine. These files contain valuable information related to machining processes. In this context, the camera device played a crucial role in capturing relevant data from these interpreted files. Subsequently, this information was relayed back to the server cloud, which served as the foundation for the IoT-based application rooted in the principles of STEP-NC. In essence, this study's achievement was the successful development of a smart system that harnessed the power of IoT to integrate machine vision inspections within the realm of CNC machining.

## Developing computer vision algorithms

The SMVI system's proficiency in accurately measuring holes within the workpiece's surface feature hinges upon the selection of appropriate algorithms. These algorithms, tailored to detect and characterize the unique attributes of holes, form the backbone of the measurement process. For instance, edge detection algorithms can meticulously outline the hole's contours, enabling precise quantification. The Canny algorithm was the best, most successful, and most commonly used for detecting edges since it has various parameters that can influence the algorithm's speed and efficacy in edge detection. This model employs the Hough transform algorithm to detect round circles. It isolates features of specific shapes within images, using extended Hough transforms when simple analytic representations are impossible. Basic Hough transforms are most commonly used for regular shapes, lines, and circles. Therefore, innovative methods that combine multiple algorithms, such as edge detection, can improve the accuracy and robustness of hole measurement. This ensures that algorithms align with SMVI system objectives for high-quality results by balancing algorithmic capabilities, computational resources, and workpiece surface complexities.

The following stages may be used to summarise the image processing methods that were employed in this study: Definition of an algorithm for processing images. Require: acquired by the machine vision system are RGB images. Verify The size of the workpiece's circle and the reference point's auxiliary lengths.

### Algorthums structure

1. Transform the RGB images into grayscale format.

2. Apply image undistortion using the specified model parameters.

3. Utilize to detect and extract edges in the images with Canny operator.

4. Locate and detect the five pattern circles within the Derlin workpiece using the Circular Hough Transform method.

5. Define the main round holes based on the object model.

6. Rectify and adjust the position of all the detected circles.

7. Compute the segmented areas of the dimensions roundness within round holes.

8. Convert the data of measured circle obtained from the camera into virtualized points compatible with the IoT environment.

9. Present the obtained results on the Visual Network Computing interface for display and analysis.

Once the RGB image is obtained, a threshold value is applied to differentiate the target region from its background in color photographs. The resulting output pixel value is determined based on the corresponding input pixel value. Such operators can be utilized in tasks such as edge detection, color manipulation, and adjustments of brightness and contrast. Image quality plays a significant role in both human and machine vision, making it an important aspect of image processing. As part of the study, a comprehensive database of workpiece samples was created, consisting of approximately 330 images that showcase various surface characteristics. This dataset includes the enhanced Part 21 file ISO14649, which incorporates five roundness holes for analysis and evaluation.

### Coorinate measureming machine (CMM)

The machine tool and measurement system integration has improved. Measurement methods have improved in efficiency and accuracy due to the benefit of more sophisticated processing capabilities [43]. The Mitutoyo QM-Measure 353 and the QM-353 manual CMM device were used to test identical specimens or workpieces. The roundness of the holes in the DELRIN workpiece was measured using the CMM measurement procedure. The results obtained from the MITUTOYO QM-Measure 353 are presented in (Fig 8). According to K. Swanson et al. [44], the MITUTOYO QM-Measure 353 is a highly precise tool for measuring roundness holes. Its specifications include a measuring range of 20" in the X-axis, 12" in the Y-axis, and an operating envelope of 20" x 12" x 12". The resolution of the device is 0.00002", with a maximum work height of 16.14" and a maximum work weight of 66 lbs. The overall dimensions of the device are L = 33", W = 35", and H = 88".

The positions in each circle are designated in (Table 3) as A, B, and C for holes with diameters of 22, 22, 22, and 22 mm, respectively. The MITUTOYO QM-Measure Coordinate Measuring Machine (CMM) uses a contact probe to detect measurements of the surface characteristic of each hole's roundness.

### Results and discussion

(Table 4) explains the surface feature diameter of case study envisioned with the four roundness circles of the absolute model of Example 1 Part21 file based ISO14649, developed using

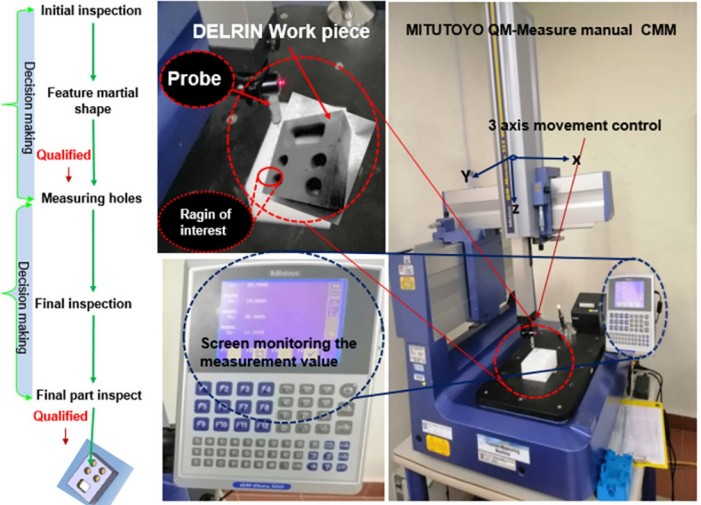

**Fig 8. Flow chart of MITUTOYO device for contact measurements of CMM.**

the 3SMVI vision system. Likewise, (Table 5) shows the points identified by probing the measurements gained of the surface feature of four roundness holes of the workpiece developed by MITUTOYO QM-Measure CMM.

As a result, (Fig 9) represents the data diameters in each point consistent and anticipated with the varied angles ranging from 0° degrees to 360 degrees of angularity. In addition, it has shown the data in red color decrease as the minimum value in three measurements. As a result, (Fig 10) demonstrated the data diameters in each point consistent and predictable with the varied points such as (A, B and C), (a, b and c). In addition, it showed the data in red color as the minimum value measurements intended to be the best value than vision system. However, the contact approach, while capable of producing precise results, the data acquired is more time-consuming than the 3SMVI system. Thus, the critical issue with contact type is wear and tear between the workpiece and the measuring probe.

Table 3. Measurement value of case study in 4 holes diameter from CMM for mild Delrin specimen.

| Image No. | Points | Hole 1 Dia. 22 mm | Hole 2 Dia. 22 mm | Hole 3 Dia. 22 mm | Hole 4 Dia. 22 mm |
|---|---|---|---|---|---|
| 1 | A, B, C | 22.0685 | 22.0087 | 22.0349 | 22.0068 |
| 2 | A, C, B | 22.0673 | 22.0095 | 22.0374 | 22.0072 |
| 3 | B, A, C | 22.0665 | 22.0092 | 22.0391 | 22.0079 |
| 4 | B, C, A | 22.0684 | 22.0052 | 22.0346 | 22.0062 |
| 5 | C, A, B | 22.0652 | 22.0061 | 22.0378 | 22.0075 |
| 6 | C, B, A | 22.0681 | 22.0039 | 22.0385 | 22.0085 |
| 7 | a, b, c | 22.0657 | 22.0047 | 22.0364 | 22.0082 |
| 8 | a, c, b | 22.0674 | 22.0071 | 22.0358 | 22.0077 |
| 9 | b, a, c | 22.0651 | 22.0064 | 22.0366 | 22.0061 |
| 10 | b, c, a | 22.0679 | 22.0063 | 22.0356 | 22.0089 |
| 11 | c, a, b | 22.0668 | 22.0041 | 22.0351 | 22.0081 |
| 12 | c, b, a | 22.0682 | 22.0054 | 22.0367 | 22.0092 |

**Table 4. Diameter values based on MZC for mild Delrin specimen in vision system.**

| Image No. | Angle in Degree | Mild Delrin Hole 1 Dia. | Mild Delrin Hole 2 Dia. | Mild Delrin Hole 3 Dia. | Mild Delrin Hole 4 Dia. |
|---|---|---|---|---|---|
| | | 22mm | 22 mm | 22 mm | 22 mm |
| 1 | 0˚ | 22.2693 | **22.1297** | 21.9897 | 21.9432 |
| 2 | 31˚ | 22.2473 | 22.1209 | **21.9797** | 21.9392 |
| 3 | 62˚ | 22.2695 | 22.1211 | 21.9808 | 21.9421 |
| 4 | 93˚ | 22.2581 | 22.1231 | 21.9801 | 21.9399 |
| 5 | 124˚ | **22.1902** | 22.1223 | 21.9808 | 21.9447 |
| 6 | 155˚ | 22.2491 | 22.1219 | 21.9891 | 21.9452 |
| 7 | 186˚ | 22.2506 | 22.1191 | **21.9756** | 21.9391 |
| 8 | 217˚ | 22.2479 | 22.1276 | 21.9822 | 21.9389 |
| 9 | 248˚ | 22.2459 | 22.1216 | 21.9814 | **21.9385** |
| 10 | 279˚ | 22.2397 | 22.1203 | 21.9823 | 21.9432 |
| 11 | 310˚ | 22.2689 | 22.1201 | 21.9799 | 21.9422 |
| 12 | 360˚ | **22.2992** | **22.1199** | 21.9805 | **21.9498** |
| | Rcmax | **22.2992** | **22.1297** | **21.9897** | **21.9498** |
| | Rcmin | **22.1902** | **22.1191** | **21.9756** | **21.9385** |
| | Roundness Circle Hole Error in mm = Rmax—Rmin | **10.9μm** | **10.6μm** | **14.1μm** | **11.3μm** |

## Inspection measurement comparison and discussion for a case study using the 3SMVI system and a CMM

(Fig 11) describes the accurate measurement data of four roundness circles in Case Study, comparing the outcome between contact and non-contact approaches. Hence, the dimension of the roundness circles shows the data from the contact approach has precise value in holes 1 and 2. Moreover, the data obtained from non-contact show highly precise that contact in roundness circles of hole 3 and hole 4. However, the data of roundness circles have as illustrated separately based on each hole, as shown in (Fig 12).

**Table 5. Diameter values based on MZC for mild Delrin model from contact approach (CMM).**

| No. of Point. | Points measuring | Mild Delrin Hole 1 Dia. | Mild Delrin Hole 2 Dia. | Mild Delrin Hole 3 Dia. | Mild Delrin Hole 4 Dia. |
|---|---|---|---|---|---|
| | | 22 mm | 22 mm | 22 mm | 22 mm |
| 3 | A,B,C | 22.0685 | 22.0087 | 22.0349 | 22.0068 |
| 3 | A,C,B | 22.0673 | 22.0095 | 22.0374 | 22.0072 |
| 3 | B,A,C | 22.0665 | 22.0092 | 22.0391 | 22.0079 |
| 3 | B,C,A | 22.0684 | 22.0052 | 22.0346 | 22.0062 |
| 3 | C,A,B | 22.0652 | 22.0061 | 22.0378 | 22.0075 |
| 3 | C,B,A | 22.0681 | 22.0039 | 22.0385 | 22.0085 |
| 3 | a, b, c | 22.0657 | 22.0047 | 22.0364 | 22.0082 |
| 3 | a, c, b | 22.0674 | 22.0071 | 22.0358 | 22.0077 |
| 3 | b, a, c | 22.0651 | 22.0064 | 22.0366 | 22.0061 |
| 3 | b, c, a | 22.0679 | 22.0063 | 22.0356 | 22.0089 |
| 3 | c, a, b | 22.0668 | 22.0041 | 22.0351 | 22.0081 |
| 3 | c, b, a | 22.0682 | 22.0054 | 22.0367 | 22.0092 |
| | Rcmax | **22.0685** | **22.0095** | **22.0391** | **22.0092** |
| | Rcmin | **22.0651** | **22.0039** | **22.0346** | **22.0061** |
| | Circle Hole Error in mm = Rcmax—Rcmin | 3.4μm | 5.6μm | 4.5μm | 3.1μm |

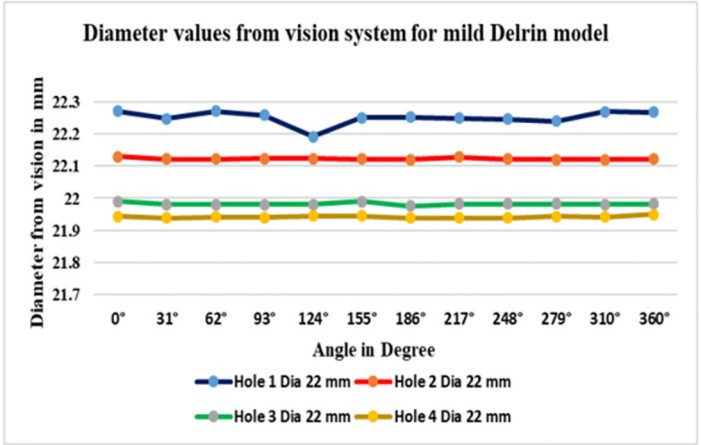

**Fig 9. Non-contact measurement data in 3SMVI for mild Delrin.**

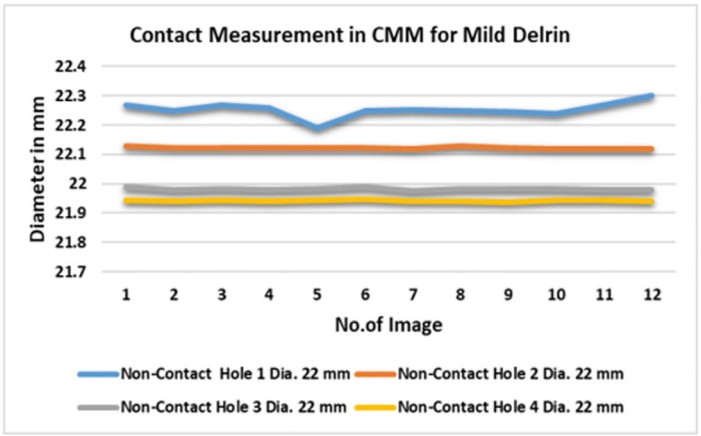

**Fig 10. Contact measurement data in CMM for mild Delrin.**

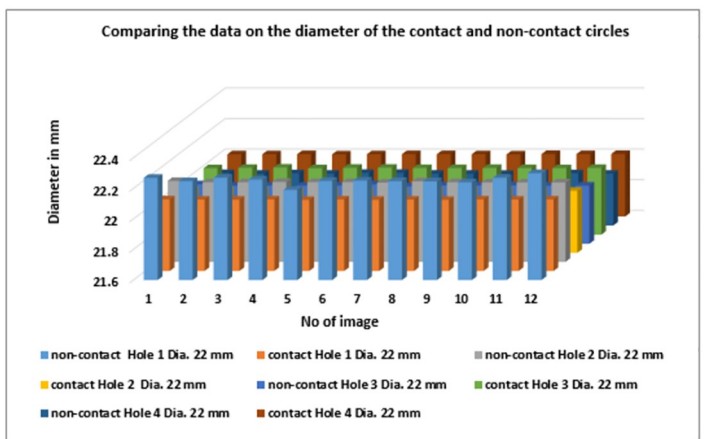

**Fig 11. Comparing the data on the diameter of the contact and non-contact circles.**

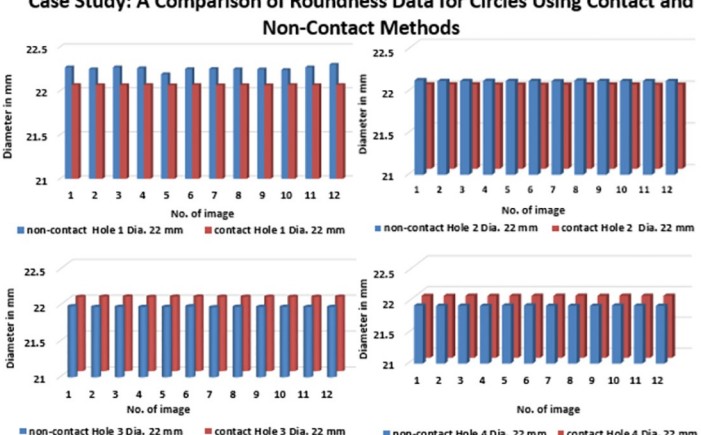

**Fig 12. A comparison between non-contact and contact measurement values for each hole in a mild Delrin workpiece in a specific case study.**

**Table 6. Comparison of diameters measured by contact and non-contact MZC method for Delrin workpiece.**

| 22mm | | 22mm | | 22mm | | 22mm | |
|---|---|---|---|---|---|---|---|
| Contact | Non-Contact | Contact | Non-Contact | Contact | Non-Contact | Contact | Non-Contact |
| **22.0651** | **22.1902** | **22.0039** | **22.1191** | **22.0346** | **21.9756** | **22.0061** | **21.9385** |
| 22.0652 | 22.2397 | 22.0041 | 22.1199 | 22.0349 | 21.9797 | 22.0071 | 21.9389 |
| 22.0657 | 22.2459 | 22.0047 | 22.1201 | 22.0351 | 21.9799 | 22.0072 | 21.9391 |

**Table 7. The average circle hole errors for each hole in a mild Delrin workpiece, comparing the values obtained through contact and non-contact measurements.**

| Diameter in mm | Hole circle error in μm from non-contact approach | Hole circle error in μm from contact approach | Difference in μm |
|---|---|---|---|
| 22 | 10.9μm | 3.4μm | 7.5 |
| 22 | 10.6μm | 5.6μm | 5 |
| 22 | 14.1μm | 4.5μm | 9.6 |
| 22 | 11.3μm | 3.1μm | 8.2 |

(Table 6) illustrates the comparison of the best data of roundness measurement within three points based on the method of MZC for contact and non-contact approaches. Therefore, the minimum measure of roundness circles is addressed in green colour.

Consequently, (Table 7) described the total errors between 3SMVI and CMM measurements based on the minimum circle zone of Case Study. However, the maximum non-contact errors were 14.1-micron meters in hole 3, while the maximum contact errors were 5.6-micron meters in hole 2. On the other hand, the smallest error measurement for the contact approach is 3.1 microns, but the minimum value of non-contact errors is 10.6-micron meters in hole 2. Thus, these approaches specify that the value contact approach was precise and accurate due to their small error to the actual value.

(Fig 13) shows the average circle hole error of both contact and non-contact measurements for mild DELRIN workpiece in the case study for four roundness circles. That acquired from the value of roundness circle dimensions the measure from each section and observed that the highest error value and the lowest error value measured in microns. Nevertheless, (Fig 14)

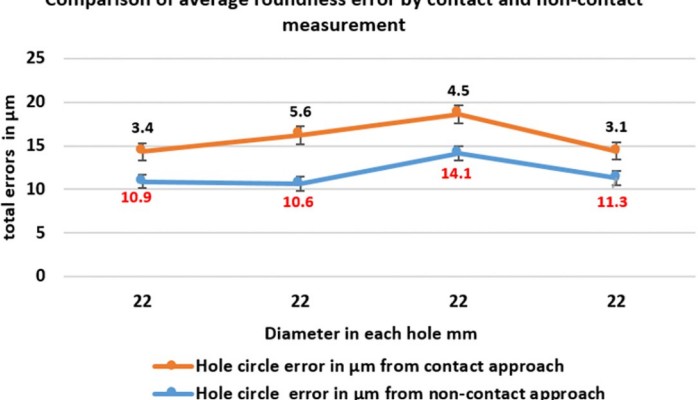

**Fig 13. The comparison graph of the average circle errors obtained through both contact and non-contact approaches using the (MZC) method.**

provides a detailed analysis of the differential errors observed in a mild DELRIN workpiece when comparing the non-contact measurements from the 3SMVI system and the contact measurements obtained from a Coordinate Measuring Machine (CMM). The graph shows discrepancies in the data ranging from 5μm to 10μm.

## Conclusions and future work

Metrology plays a crucial role in the manufacturing industry based on Industrial 4.0. In this era of advanced manufacturing, which emphasizes automation, customization, and globalization, precise and accurate measurement is vital for quality control and process optimization. Metrology holds significant importance in the modern manufacturing industry, and its significance is expected to grow as the industry evolves and adopts new technologies and processes.

Thus, this article highlights the need for modern industrial devices to communicate through local (edge) and cloud computing servers. To enhance data transmission performance to and from cloud servers, the Industrial IoT protocol aims to leverage the MQTT server. Therefore, MQTT's flexibility and protocol-agnostic nature make it adept at handling various sensor types, data formats, and communication protocols. By decoupling producers and consumers through its publish-subscribe model, MQTT allows different sensors to communicate

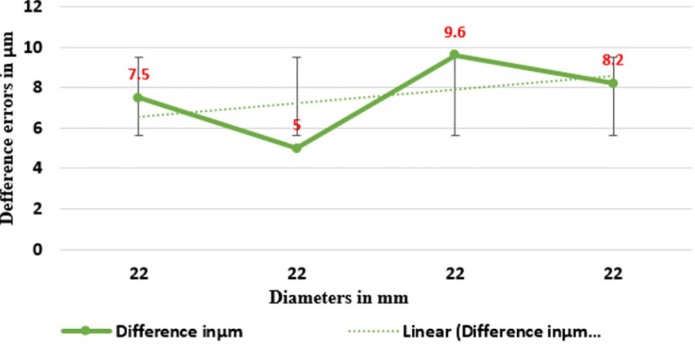

**Fig 14. The difference errors between 3SMVI and CMM in micron meter.**

seamlessly, reducing the need for extensive protocol translation or adaptation. MQTT aligns with the principles of Industry 4.0 and offers substantial advantages for the manufacturing industry, ranging from process improvements to cost savings. This research contributes significantly to the advancement of the 3SMVI system, which utilizes IoT architecture to measure the roundness of holes in Delrin workpieces. Additionally, the MQTT protocol facilitates machine-to-machine communication, enables identification and examination of hole diameters on a cloud platform, and allows interoperable data collection of surface properties of workpieces. These innovation platforms provide an advanced approach for achieving seamless coordination between machining, inspection, and cloud-based applications by successfully integrating the IoT's potential with interpreted STEP-NC files.

The study introduces an improved non-contact method that is non-invasive and specifically designed to verify the roundness of circular holes. The minimal zone algorithm, employed to calculate roundness error, proves efficient in locating discrete data points on a uniform circle, making it suitable for continuous quality inspection. The Minimum Zone Circle (MZC) method is employed to adhere to ISO and ANSI standards, providing more precise values for prismatic milling parts compared to other methods. The inspection results obtained from the 3SMVI system are compared to those obtained from traditional measuring machines like the CMM. Moreover, the study examines the difference in error between non-contact and contact measurements. By utilizing machine vision inspection and an intelligent system-based STEP-NC file interpretation, the inspection accuracy ranges from ± 9μm to ± 14 μm.

In future work, the study plans to improve image quality, update algorithms, and use cloud computing protocols for data transmission. It will integrate deep learning into CNNs for surface feature detection, and use Tensor Flow-assisted real-time image categorization for accurate feature detection. Additionally, to delve deeper into latency analysis through MQTT and its implications for circularity measurements. Thus, a more detailed investigation into this area could provide valuable insights into the protocol's performance in real-time scenarios. This study has the potential to benefit many industries, including product quality, inspection, time capturing data, and dimensional analysis.

## Supporting information

**S1 File.**
(DOCX)

## Acknowledgments

The authors are thankful to the Department of Industrial Engineering, College of Engineering at Alfaisal University, in Saudi Arabia, and the faculty of Mechanical and Manufacturing Engineering at the University Tun Hussien ONN Malaysia (UTHM) for their kind assistance in our studies.. Additionally, the authors wish to extend their sincere appreciation to the Institute for Integrated Engineering, and the facilities of the Sustainable Polymer Engineering, Advanced Manufacturing, and Material Centre (SPEN-AMMC).

## Author Contributions

**Conceptualization:** Sami Al-Alimi, Kamran Latif, Hamood Alshalabi, Safwan Sadeq.

**Data curation:** Yazid Saif, Djamal Hissein Didane, Aini Zuhra Abdul Kadir.

**Formal analysis:** Yazid Saif, Anika Zafiah M. Rus, Kamran Latif.

**Funding acquisition:** Yusri Yusof, Anika Zafiah M. Rus.

**Investigation:** Yazid Saif, Yusri Yusof, Sobhi Mejjaouli, Djamal Hissein Didane.

**Methodology:** Yazid Saif, Yusri Yusof, Atef M. Ghaleb, Sobhi Mejjaouli, Kamran Latif.

**Resources:** Yazid Saif, Sami Al-Alimi, Djamal Hissein Didane, Hamood Alshalabi, Safwan Sadeq.

**Software:** Yazid Saif.

**Supervision:** Yusri Yusof, Anika Zafiah M. Rus.

**Visualization:** Atef M. Ghaleb, Sobhi Mejjaouli.

**Writing – original draft:** Yazid Saif.

**Writing – review & editing:** Yusri Yusof, Anika Zafiah M. Rus, Atef M. Ghaleb, Sami Al-Alimi, Aini Zuhra Abdul Kadir.

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
