## [Decision Letter · Decision Letter 0]

21 Aug 2023

PONE-D-23-19982MQTT Protocol for Circularity Measurements in Industry 4.0-Based Manufacturing Metrology: A Case StudyPLOS ONE

Dear Dr. Saif,

Thank you for submitting your manuscript to PLOS ONE. After careful consideration, we feel that it has merit but does not fully meet PLOS ONE’s publication criteria as it currently stands. Therefore, we invite you to submit a revised version of the manuscript that addresses the points raised during the review process.

We look forward to receiving your revised manuscript.

Kind regards,

Nadeem Sarwar

Academic Editor

PLOS ONE

Journal Requirements:

   "The authors would like to express their deepest appreciation to the Ministry of Higher Education (MOHE) through the fundamental Grant Scheme (FRGS/1/2020/STG01/UTHM/02/2, Malaysia and Alfaisal University, Saudi Arabia for funding this project. Additional support in terms of facilities was also provided by Sustainable Polymer Engineering, Advanced Manufacturing and Material Centre (SPEN-AMMC)."

Reviewers' comments:

Reviewer's Responses to Questions

**Comments to the Author**

1. Is the manuscript technically sound, and do the data support the conclusions?

Reviewer #1: Yes

Reviewer #2: Yes

Reviewer #3: Yes

Reviewer #4: Yes

Reviewer #5: Yes

2. Has the statistical analysis been performed appropriately and rigorously? 

Reviewer #1: Yes

Reviewer #2: N/A

Reviewer #3: Yes

Reviewer #4: Yes

Reviewer #5: Yes

3. Have the authors made all data underlying the findings in their manuscript fully available?

Reviewer #1: Yes

Reviewer #2: Yes

Reviewer #3: Yes

Reviewer #4: Yes

Reviewer #5: Yes

4. Is the manuscript presented in an intelligible fashion and written in standard English?

Reviewer #1: Yes

Reviewer #2: Yes

Reviewer #3: Yes

Reviewer #4: Yes

Reviewer #5: Yes

5. Review Comments to the Author

Reviewer #1: The case study focuses on circularity measurement using MQTT protocol in Industry 4.0 based on Manufacturing Meterology

1. Kindly correct the heading (Subdivision - 6) Coordinate Measuring Machine (CMM)

2. 5.1 heading correction (Structure)

Reviewer #2: 1. Appendices are missing

2. It is mentioned in section 3.2.1. that the CAD model identifies surface areas and emphasizes holes for inspection. However, it doesn't detail how the CAD model selection and inspection strategy were determined. Providing insights into the reasoning behind the chosen areas and features for inspection would enhance the credibility of the design.

3. he provided machining parameters are valuable for understanding the setup, but there's no discussion about the rationale behind selecting these specific parameters. Explaining why the spindle speed, feed rate, depth of cut, and cutting speed were chosen could enhance the reader's understanding of the machining process.

4. Various hardware components of the 3SMVI system, including a webcam, power supply, internet adapter, lighting system, step motor, personal computer, and driver are mentioned. However, specific details about the selection criteria for these components and how they integrate cohesively to create the vision inspection system could be valuable.

5. Mentioning the webcam's capability to capture surface feature images up to 5 megapixels and its resolution is helpful. However, further context on why this particular webcam was chosen, considering the inspection requirements and performance expectations, would provide a clearer understanding of the hardware decision-making process.

6. The mention of IoT integration through elements like FileZilla, Raspberry Pi, LED lighting, and Wi-Fi router is intriguing. Elaborating on how these components work together and contribute to the system's IoT capabilities would enhance comprehension.

7. Mentioning the use of OpenCV to develop inspection modules for detection and measurement is important. Providing examples of the types of measurements and analyses performed by these modules would give readers a clearer picture of the system's capabilities.

8. It's important to discuss how the chosen algorithms and pre-processing steps affect the overall quality of the results. How do these steps contribute to improving the accuracy of measurements or enhancing the system's capabilities? This connection should be explicitly highlighted.

9. Consider discussing the performance of different algorithms if applicable. Are there specific algorithms that outperformed others in certain scenarios? Exploring these comparative aspects can provide insights into the decision-making process behind algorithm selection.

Reviewer #3: The present article entitled "MQTT Protocol for Circularity Measurements in Industry 4.0-Based Manufacturing Metrology: A Case Study" investigates the use of the MQTT protocol to enhance the performance of circularity measurement data transmission between cloud servers and round-hole data sources through Open CV. The system implements the MZC problem solution technique to improve roundness measurement. A workpiece with circular features has been embedded in the IoT- enabled vision machine system. Overall article is written well and have merit for publication.

Reviewer #4: The title focuses on the MQTT protocol while the work and the novelty emphasises on the image processing techniques proposed for the measurements. Either the title should be revised to reflect the work better or the details of the MQTT protocol used should be added to the manuscript.

Reviewer #5: The paper presents an interesting case study on using the MQTT protocol for circularity measurements in Industry 4.0-based manufacturing metrology. However, the methodology section lacks details on the specific instruments and devices used for the circularity measurements. Providing information on the measurement equipment, accuracy, and calibration procedures would enhance the robustness of the study.

It's important to clarify the sample size and variability in the experiments. Elaborate on how the data were collected, including the frequency of measurements and the conditions under which they were conducted.

While the paper introduces the use of MQTT for data communication, it falls short in explaining the technical aspects of implementing MQTT in the manufacturing environment. More details on how MQTT was integrated with the existing systems, including the configuration of topics, quality of service levels, and message retention policies, would greatly benefit readers looking to replicate the study.

The paper mentions the efficiency of MQTT in handling real-time data transmission. However, a deeper analysis of the latency introduced by the protocol and its impact on the timeliness of circularity measurements should be included.

To provide a well-rounded perspective, it's recommended to compare the proposed MQTT protocol with other communication protocols commonly used in Industry 4.0 scenarios, such as CoAP or AMQP. Discuss the pros and cons of each protocol in terms of data throughput, reliability, and suitability for different manufacturing scenarios.

MQTT's security features, such as authentication and encryption, play a critical role in industrial applications. The paper should address the security measures implemented to protect the transmitted circularity data, including user authentication, message encryption, and secure communication channels.

Industry 4.0 environments often involve heterogeneous devices and systems. Discuss how the proposed MQTT solution addresses interoperability challenges, especially when dealing with various sensor types, data formats, and communication protocols.

Consider expanding on the scalability of the MQTT solution. How does it handle an increasing number of sensors and devices in a large manufacturing setup? Are there any limitations or potential bottlenecks?

The paper concludes with a promising case study, but it lacks a discussion on the practical implications for the manufacturing industry. Consider elaborating on the potential benefits of adopting the proposed MQTT protocol for circularity measurements, including improved process control, reduced downtime, and enhanced data-driven decision-making.

Suggest avenues for future research, such as exploring how the protocol could be extended to address challenges related to different types of metrology measurements or integration with higher-level manufacturing systems.

6. PLOS authors have the option to publish the peer review history of their article (what does this mean?). If published, this will include your full peer review and any attached files.

Reviewer #1: **Yes: **Arvind C

Reviewer #2: No

Reviewer #3: No

Reviewer #4: **Yes: **Rizwan Aslam BUTT

Reviewer #5: **Yes: **Muhammad Aamir

---

## [Author Response · Author response to Decision Letter 0]

21 Sep 2023

Response to Reviewer 1

Questions Comments Line No.630, 607

The case study focuses on circularity measurement using MQTT protocol in Industry 4.0 based on Manufacturing Metrology

1. Kindly correct the heading (Subdivision - 6) Coordinate Measuring Machine (CMM)

2. 5.1 heading correction (Structure) Thank you for the acknowledgment and the comment. We really appreciate your comment. 

We had edited and changed the heading from “Coordinate measureming Machine by MITUTOYO Device” to “Coorinate Measureming Machine (CMM) (highlighted in red color). 

We have changes the word from “stracture” to structure were (highlighted in red color). 

630, 607

Response to Reviewer 2

Questions Comments Line No.

1. Appendices are missing

2. It is mentioned in section 3.2.1. that the CAD model identifies surface areas and emphasizes holes for inspection. However, it doesn't detail how the CAD model selection and inspection strategy were determined. Providing insights into the reasoning behind the chosen areas and features for inspection would enhance the credibility of the design.

3. he provided machining parameters are valuable for understanding the setup, but there's no discussion about the rationale behind selecting these specific parameters. Explaining why the spindle speed, feed rate, depth of cut, and cutting speed were chosen could enhance the reader's understanding of the machining process.

4. Various hardware components of the 3SMVI system, including a webcam, power supply, internet adapter, lighting system, step motor, personal computer, and driver are mentioned. However, specific details about the selection criteria for these components and how they integrate cohesively to create the vision inspection system could be valuable.

5. Mentioning the webcam's capability to capture surface feature images up to 5 megapixels and its resolution is helpful. However, further context on why this particular webcam was chosen, considering the inspection requirements and performance expectations, would provide a clearer understanding of the hardware decision-making process.

6. The mention of IoT integration through elements like FileZilla, Raspberry Pi, LED lighting, and Wi-Fi router is intriguing. Elaborating on how these components work together and contribute to the system's IoT capabilities would enhance comprehension.

7. Mentioning the use of OpenCV to develop inspection modules for detection and measurement is important. Providing examples of the types of measurements and analyses performed by these modules would give readers a clearer picture of the system's capabilities.

8. It's important to discuss how the chosen algorithms and pre-processing steps affect the overall quality of the results. How do these steps contribute to improving the accuracy of measurements or enhancing the system's capabilities? This connection should be explicitly highlighted.

9. Consider discussing the performance of different algorithms if applicable. Are there specific algorithms that outperformed others in certain scenarios? Exploring these comparative aspects can provide insights into the decision-making process behind algorithm selection. 

We appreciate your fruitful comments. We have added the missing Appendices for A and B in pdf file as supplementary. 

Thanks for your comment. We had added some sentences. “This study utilizes a STEP-NC File 21 compliant reference model to prioritize inspection holes and surface area identification in a CAD model. This transparent documentation strengthens the design's validity and aligns with STEP-NC principles, enhancing accuracy and quality assurance in the CNC milling machine model.”

we appreciate your comments. We added these sentences to the manuscript. “The guidelines for developing the STEP-NC code of standardization for milling CNC machines are exactly what is identified in this machine's parameters. However, this data could be applied by a PROLIGHT 100 CNC milling machine to drill the four holes in the Delrin workpiece in accordance with the standards to obtain better surface features after the product's manufacturing is succeeded.” (highlighted in red color)

Thanks for your fruitful comment. We added some sentences in detail. The integration of the 3SMVI system has been added in the new section 4.3, that explains the process of 3SMVI integration. (highlighted in red color)

Thanks for your comment. The chosen camera was due to the availability inside Advanced Material and Manufacturing Centre Lab (AMMC) in UTHM, as detailed with specification camera 640x480 resolutions up to 5 Megapixels. 

We appreciate your comment. We had elaborated some sentences that addressed the on how the FileZilla, Raspberry pi, LED and Wi-Fi works together cloudily to relies on IoT environment. (highlighted in red color).

Thanks for your great comment. We had previously provided, as shown in figure 7. measuring the surface feature of four holes of Delrin's workpiece based on the OpenCV platform as an adequate example in this study and analyzing the data through multi-algorithms implemented to capture the size of the hole. (highlighted in red color)

Thanks for the fruitful comment. We had implement minimum circle zone algorithm that utilized to identify the errors roundness images for twelve points. Furthermore, the canny edges was utilized and Hough transform was utilized. (highlighted in red color)

Thanks for your comments. We added a sentences to clarify the best algorithms that had implemented in 3SMVI system. (highlighted in red color)

223-243

250-259

564-586

305-328

335-353

552-559

588-601

588-598, 608-612

Response to Reviewer 3

Questions Comments 

The present article entitled "MQTT Protocol for Circularity Measurements in Industry 4.0-Based Manufacturing Metrology: A Case Study" investigates the use of the MQTT protocol to enhance the performance of circularity measurement data transmission between cloud servers and round-hole data sources through Open CV. The system implements the MZC problem solution technique to improve roundness measurement. A workpiece with circular features has been embedded in the IoT- enabled vision machine system. Overall article is written well and have merit for publication. Thanks for your fruitful comments. We appreciate your valuables comment. As a way of improving roundness measurement, the system uses the MZC problem-solving approach. The IoT-enabled vision machine system has a workpiece with circular features embedded in it. These innovation platforms provide a fresh method for achieving seamless coordination between machining, inspection, and cloud-based applications by effectively combining interpreted STEP-NC files with IoT potential.

Response to Reviewer 4

Questions Comments Line No.

The title focuses on the MQTT protocol while the work and the novelty emphasises on the image processing techniques proposed for the measurements. Either the title should be revised to reflect the work better or the details of the MQTT protocol used should be added to the manuscript.

 Thanks for your fruitful comment. We have revised and changed the title from “MQTT Protocol for Circularity Measurements in Industry 4.0-Based Manufacturing Metrology: A Case Study” to “Implementing Circularity Measurements in Industry 4.0-Based Manufacturing Metrology Using MQTT Protocol and Open CV: A Case Study”. Furthermore, we had added intensive details regarding the MQTT protocols on how the data transmitted through the cloud using some important steps with Real virtual network computing.

1-2

423-438, 

453-492

Response to Reviewer 5

Questions Comments Line No.

The paper presents an interesting case study on using the MQTT protocol for circularity measurements in Industry 4.0-based manufacturing metrology. However, the methodology section lacks details on the specific instruments and devices used for the circularity measurements. Providing information on the measurement equipment, accuracy, and calibration procedures would enhance the robustness of the study.It's important to clarify the sample size and variability in the experiments. Elaborate on how the data were collected, including the frequency of measurements and the conditions under which they were conducted.

 Thanks for your interesting comments. We appreciate your point regarding the MQTT luck in methodology’s. However, the 3SMVI interface platform is designed to determine the pixel size in both the x and y axes by taking into account several crucial parameters. These parameters include the actual size of the expanded model, which is derived from the STEP-NC Example 1 Part 21 file. Furthermore, it's important to note that the program considers the camera system calibration. This calibration is instrumental in establishing the relationship between the camera's imaging properties and the real-world dimensions it captures. In particular, it addresses the region of interest (ROI), which defines the portion of the image that is relevant for analysis. By incorporating camera calibration and knowledge of the ROI, the program ensures accurate and precise pixel-to-real-world unit conversion, which is vital for various applications in image analysis and measurement. (Highlighted in red color)

272-292

While the paper introduces the use of MQTT for data communication, it falls short in explaining the technical aspects of implementing MQTT in the manufacturing environment. More details on how MQTT was integrated with the existing systems, including the configuration of topics, quality of service levels, and message retention policies, would greatly benefit readers looking to replicate the study. 

 We appreciate your comments. We had added paragraphs in section 3.4 based on the steps of how to install MQTT inside python through Open CV. 

We addressed the sequence of implementing the MQTT in to 3SMVIsystem through Open CV platform. 

393-415,

453-492

The paper mentions the efficiency of MQTT in handling real-time data transmission. However, a deeper analysis of the latency introduced by the protocol and its impact on the timeliness of circularity measurements should be included. Thank you for your kindness comment. Regarding the latency analysis of MQTT in the context of circularity measurements. In this paper, our primary focus was to highlight the measurements of the holes through Open CV based on the cloud server within MQTT protocols that can transmissions the data in real-time within the context of Industry 4.0-based manufacturing metrology. 

These statement shows the efficiency of the CMM as contact measuring related to improved non-contact method that is non-invasive and specifically designed to verify the roundness of circular holes. The minimal zone algorithm, employed to calculate roundness error, proves efficient in locating discrete data points on a uniform circle, making it suitable for continuous quality inspection. -----

118,

453-457,

791-801

To provide a well-rounded perspective, it's recommended to compare the proposed MQTT protocol with other communication protocols commonly used in Industry 4.0 scenarios, such as CoAP or AMQP. Discuss the pros and cons of each protocol in terms of data throughput, reliability, and suitability for different manufacturing scenarios. Thanks for your great comment. We had added a sentences with references [39-41] supporting the MQTT protocol was the best for this study due to the limit of wireless. In comparison to CoAP, MQTT supports both wired and wireless connections, has a smaller memory footprint than AMQP, and offers a greater variety of Quality of Service (QoS) parameter options that guarantee message delivery for critical applications or transmission only when necessary to reduce network bandwidth and energy consumption. 423-438

MQTT's security features, such as authentication and encryption, play a critical role in industrial applications. The paper should address the security measures implemented to protect the transmitted circularity data, including user authentication, message encryption, and secure communication channels. We appreciate your comment. We had addressed the security of our communication protocol based on IP address with portable internet. By controlling the transmission data within the area of our lab which used within TP-Link router, Laptop, Smart Phone, and PC. (Highlighted in red color) 

453-492

524-525,

Industry 4.0 environments often involve heterogeneous devices and systems. Discuss how the proposed MQTT solution addresses interoperability challenges, especially when dealing with various sensor types, data formats, and communication protocols. Consider expanding on the scalability of the MQTT solution. How does it handle an increasing number of sensors and devices in a large manufacturing setup? Are there any limitations or potential bottlenecks? Thank you for your insightful comments and questions regarding the MQTT solution's capabilities in addressing interoperability challenges. 

• The proposed MQTT solution is crucial in Industry 4.0 environments due to its flexibility and protocol-agnostic nature. It allows seamless communication between sensors and reduces protocol translation. MQTT's lightweight, asynchronous messaging model is ideal for large-scale deployments. However, potential bottlenecks like hardware resources and network bandwidth need careful planning and resource allocation.

• The scalability of an MQTT implementation depends on factors like the choice of MQTT broker and the network architecture. 287-292,

The paper concludes with a promising case study, but it lacks a discussion on the practical implications for the manufacturing industry. Consider elaborating on the potential benefits of adopting the proposed MQTT protocol for circularity measurements, including improved process control, reduced downtime, and enhanced data-driven decision-making. Thank you for your valuable comment. regarding the practical implications of our research. We acknowledge the importance of discussing how the adoption of the MQTT protocol for circularity measurements can benefit the manufacturing industry. 777-783

788-790

Suggest avenues for future research, such as exploring how the protocol could be extended to address challenges related to different types of metrology measurements or integration with higher-level manufacturing systems. Thanks for you fruitful suggested. We agree that latency analysis is a critical aspect, especially in timeliness-sensitive applications. While it wasn't the primary scope of this paper, we recognize its importance. In future research, we plan to delve deeper into latency analysis and its implications on circularity measurements. We believe that a more detailed investigation into this area could provide valuable insights into the protocol's performance in real-time scenarios. 802-809

---

## [Decision Letter · Decision Letter 1]

29 Sep 2023

Implementing Circularity Measurements in Industry 4.0-Based Manufacturing Metrology Using MQTT Protocol and Open CV: A Case Study

PONE-D-23-19982R1

Dear Dr. Saif,

We’re pleased to inform you that your manuscript has been judged scientifically suitable for publication and will be formally accepted for publication once it meets all outstanding technical requirements.

Kind regards,

Nadeem Sarwar

Academic Editor

PLOS ONE

Additional Editor Comments (optional):

Reviewers' comments:

Reviewer's Responses to Questions

**Comments to the Author**

1. If the authors have adequately addressed your comments raised in a previous round of review and you feel that this manuscript is now acceptable for publication, you may indicate that here to bypass the “Comments to the Author” section, enter your conflict of interest statement in the “Confidential to Editor” section, and submit your "Accept" recommendation.

Reviewer #2: All comments have been addressed

Reviewer #5: All comments have been addressed

2. Is the manuscript technically sound, and do the data support the conclusions?

Reviewer #2: Yes

Reviewer #5: Yes

3. Has the statistical analysis been performed appropriately and rigorously? 

Reviewer #2: N/A

Reviewer #5: Yes

4. Have the authors made all data underlying the findings in their manuscript fully available?

Reviewer #2: Yes

Reviewer #5: Yes

5. Is the manuscript presented in an intelligible fashion and written in standard English?

Reviewer #2: Yes

Reviewer #5: Yes

6. Review Comments to the Author

Reviewer #2: The authors have successfully incorporated all the reviewer comments. Now the quality of manuscript is improved significantly. It can be accepted in its present form.

Reviewer #5: Accepted after the clearly evaluation the grammatical errors because the paper is interesting to read.

7. PLOS authors have the option to publish the peer review history of their article (what does this mean?). If published, this will include your full peer review and any attached files.

Reviewer #2: No

Reviewer #5: No

---

## [Editor Report · Acceptance letter]

5 Oct 2023

PONE-D-23-19982R1 

Implementing Circularity Measurements in Industry 4.0-Based Manufacturing Metrology Using MQTT Protocol and Open CV: A Case Study 

Dear Dr. Saif:

I'm pleased to inform you that your manuscript has been deemed suitable for publication in PLOS ONE. Congratulations! Your manuscript is now with our production department. 

Kind regards, 

on behalf of

Dr. Nadeem Sarwar 

Academic Editor

PLOS ONE